# Dietary Habits, Food Product Selection Attributes, Nutritional Status, and Depression in Middle-Aged and Older Adults with Dysphagia

**DOI:** 10.3390/nu14194045

**Published:** 2022-09-28

**Authors:** Dahyeon Ko, Jieun Oh, Soyoung Joo, Ju Yeon Park, Mi Sook Cho

**Affiliations:** 1Department of Nutritional Science and Food Management, Ewha Womans University, Seoul 03760, Korea; 2College of Science and Industry Convergence, Ewha Womans University, Seoul 03760, Korea; 3Hyundai Green Food Co., Yongin 16827, Korea

**Keywords:** dysphagia, nutritional status, depression, dietary habits

## Abstract

Dysphagia, which increases the risk of malnutrition and depression, is an important health concern. A total of 304 people aged 50 years or above (148 subjects with dysphagia and 156 non-dysphagia subjects) were recruited for this survey of dietary habits, meal product selection attributes, nutritional status, and depression. For group comparisons, chi-square tests were performed. Exploratory factor analysis was conducted for the meal product selection attributes. Correlation analyses were performed to investigate links between EAT-10 (The 10-item Eating Assessment Tool), nutrition (Nutrition Quotient/Nutrition Quotient for the Elderly, NQ/NQ-E) and depression (The Short-Form Geriatric Depression Scale for Koreans, SGDS-K). Logistic regression analysis was performed to investigate links between EAT-10, nutritional status, and depressive status. Finally, a correlation analysis and logistic regression analysis of nutritional status, depression status, and some dietary factors were performed, targeting only the responses of the dysphagia patients. The average ages were 73.79 years in the dysphagia group and 70.15 years in the non-dysphagia group, and the total average age was 71.88 years. The overall age range was 50 to 92 years. Dysphagia (EAT-10) had significant effects on malnutrition (*β* = 0.037, OR = 1.095) and depression (*β* = 0.090, OR = 1.095) (*p* < 0.001). There was a significant correlation between SGDS-K, needing help with meals, and the amount of food consumed at mealtimes (*p* < 0.01). The correlation coefficient between SGDS-K and the need for help with meals was 0.474. Dietary factors that affected depression in dysphagia patients were the increase in the need for meal assistance (*β* = 1.241, OR = 3.460, *p* < 0.001) and the amount of food eaten at mealtimes (*β* = −0.494, OR = 0.702, *p* < 0.05). Dysphagia can increase the risk of depression and malnutrition. To reduce depression in dysphagia patients, it is necessary to develop meal products that address dietary discomfort among patients with dysphagia.

## 1. Introduction

Dysphagia is defined as difficulty swallowing due to problems caused by disorders in saliva secretion, chewing, and/or aspiration defense mechanisms [1,2]. The main symptoms of dysphagia are drooling, coughing, and chewing disorders [3], which can cause life-threatening aspiration pneumonia in patients [4,5]. In addition, the consequences of dysphagia are delayed oral intake recovery [6], extended inpatient treatment, and poor quality of life [7,8]. In particular, as many middle-aged people over the age of 50 have been reported to have dysphagia, dysphagia has become an important health concern [9,10,11,12]. Dysphagia increases the risk of malnutrition, as food intake becomes difficult [13,14,15]. In addition, dysphagia can cause a loss of self-esteem and impairment in social relationships [16], further increasing depression [17,18].

Currently, clinical intervention is focused on nutritional supply [19] and rehabilitation [20] to improve the swallowing function and prevent complications such as aspiration pneumonia. If oral intake is possible, texture-modified foods (TMFs) can be used for meals by adding a thickening agent to adjust the viscosity or by finely mincing food items into a puree or mousse and softening them through water-rich cooking [21,22,23,24,25]. Changes in physical functions due to aging affect quality of life [26,27] and dietary habits, such as meal preparation and eating habits [28]. However, studies on the dietary habits and characteristics of patients with dysphagia are insufficient. Therefore, this study aims to understand the dietary habits of the middle-aged, their selection attributes of meal products, current nutritional status, and the degree of depression according to the presence or absence of dysphagia. This will enable the development of customized foods and suggestions for patients with dysphagia. Ultimately, through this study, we intend to present basic data in order to discover a way of mitigating the difficulty of aging.

## 2. Materials and Methods

### 2.1. Study Design and Subjects

This study recruited two groups of subjects: a dysphagia group and a non-dysphagia group. One hundred and forty-eight (148) dysphagia patients aged 50 years or older and 156 adults aged 50 years or older without dysphagia were asked to respond to a survey about their nutritional status, levels of depression, and dietary habits. The sample size was calculated using the G*power 3.1.9.7 program, and the minimum number of samples was 266 based on the F test, ANOVA (fixed effect size f 0.2, alpha error probability 0.05, power 0.9, and number of groups 3). Based on the minimum sample size of 266 and an estimated dropout rate of about 15% (e.g., insincere responses), the total target number of study subjects for recruitment was calculated to be 312 subjects. Study subjects were recruited by posting research subject recruitment documents on bulletin boards and online communities in hospitals and welfare centers in Seoul, Korea. The inclusion criteria for the subjects of the dysphagia group were as follows. Adults aged 50 or older with dysphagia were selected as subjects of the study if they agreed to participate in the survey used in this study. In the case of the dysphagia group, this included people who suffer during eating activities due to the decreased muscle strength of swallowing-related muscles, decreased chewing power and saliva secretion due to tooth loss, or patients with conditions such as dementia and stroke. In the case of the normal group, adults aged 50 or older without symptoms of dysphagia were selected as subjects of the study. Due to the pandemic, it was not possible to participate in the study if any of the following were applicable: abnormal health conditions, such as coughing, sore throat, headache, muscle pain, and chills on the same day; a history of having traveled abroad within the last 14 days; having had contact with an individual in self-quarantine; and not agreeing to participate in the study. A total of 304 responses were collected and 296 survey responses were analyzed, excluding three people with unreliable responses, two people who did not answer the NQ or NQ-E items, and three people who did not meet the age criterion of being 50 years old or older. Respondents were categorized based on their total scores in EAT-10, which is a self-questioning tool used to measure swallowing disabilities, with subjects who scored more than three points classified as the dysphagia group and those who scored less than three points classified as the non-dysphagia group [29]. A questionnaire was constructed by investigating and analyzing pertinent existing research, and the survey was conducted from 20 August to 26 November 2021. To ensure that the research was ethical, the study protocol was approved by the Institutional Review Board (IRB) of Ewha Womans University, Seoul, Korea (IRB No. ewha-202107-0024-02). Written informed consent to participate in the study was obtained from all subjects.

### 2.2. Measurements

Our survey was conducted using a questionnaire consisting of 10 dysphagia measurement items, 10 items to measure dietary habits, 25 items to investigate selection attributes of meal products, 15 depression measurement items, 22 nutrition measurement items, and 11 demographic characteristics. The purpose of this study was to compare nutrition, depression, and dietary factors according to the presence or absence of dysphagia. Therefore, the independent variable was the presence or absence of dysphagia, and the dependent variables were established as dietary factors, selection attributes of meal products, nutrition, and depression. The total dysphagia score was obtained using the Eating Assessment Tool (EAT-10), and the dysphagia group and the non-dysphagia group were classified. Depression was measured using the Short-Form Geriatric Depression Scale for Koreans (SGDS-K), and nutrition was measured using the Nutrition Quotient for the Elderly (NQ-E) and Nutrition Quotient (NQ).

The Eating Assessment Tool (EAT-10) is a dysphagia clinical evaluation method developed by Belafsky et al. (2008), which consists of 10 questions and enables screening through short and comprehensive items [29]. Scores in EAT-10 are calculated by the self-evaluation of dysphagia symptoms, represented as values ranging from 0 to 4 (self-reported answers of “no problems”, equating to 0 points, escalating to “severe problems”, equating to 4 points), with three questions to measure functional aspects, three questions to measure emotional aspects, and four questions to measure physical aspects of dysphagia. If the total score for all 10 questions is 3 or more out of 40 points, then the result is classified as abnormal and the subject is classified as having dysphagia [29]. The EAT-10 tool has been used in many previous studies to identify patients with dysphagia and has been utilized in studies focusing on nutritional status and quality of life in stroke patients suffering from dysphagia [30,31].

After classifying the dysphagia group and the non-dysphagia group using EAT-10, the dietary factors and meal product selection attributes according to the presence or absence of dysphagia were investigated and compared. Dietary factors included needing help with meals, the amount of food consumed at meals, duration of mealtimes, discomfort while eating, the person who prepares meals, person who buys food, and the cost of food (per month). The questionnaire referred to previous studies on the diet and meals of middle-aged and older adults in Korea [32,33,34,35].

The Nutrition Quotient for the Elderly (NQ-E) and the Nutrition Quotient (NQ) are nutrition evaluation tools for the elderly, such as the Dietary Quality Index (DQI) and Healthy Eating Index (HEI). NQ-E and NQ were developed to enhance the validity of nutritional judgment by reflecting the diet in Korea so as to meet the needs of a nutrition screening tool that reflects the characteristics and physical and social factors [36,37]. NQ-E is a dietary evaluation tool for elderly people developed by Chung (2018), which consists of a total of 19 questions in four detailed areas, including “Dietary Behavior”, “Moderation”, “Balance”, and “Diversity”. Scores are calculated for each of the four areas, and the meal quality and nutritional status are evaluated in the context of elderly eaters in Korea [38]. NQ is a tool developed by Lee (2018) that comprehensively evaluates nutritional status and meal quality for adults aged 19 to 64 [39]. Similar to NQ-E, the metric is divided into the same four areas of “Dietary Behavior”, “Moderation”, “Balance”, and “Diversity”, and consists of a total of 21 questions. Both NQ-E and NQ can be used without specialized knowledge of nutrition and with simple survey methods, as evaluation tools that reflect Korean dietary patterns and eating habits [36,37]. NQ and NQ-E are authorized evaluation tools developed with the support of the Ministry of Food and Drug Safety in Korea. In this study, nutritional scores were calculated using NQ and NQ-E, and the age groups of the subjects (50 to 65 and 65 or older) were classified as having upper, middle, or lower nutritional status based on the scores.

The body mass index (BMI) was calculated using the measured height and weight responses as an expression of weight (kg)/height (m^2^), and a BMI of <18.5 was classified as a low weight, 18.5 ≤ BMI < 23 was normal, 23 ≤ BMI < 25 was classified as overweight, and 25 < BMI was classified as obese [40].

The Short-Form Geriatric Depression Scale (SGDS) is a short version of the Geriatric Depression Scale (GDS), a measure of depression in elderly people developed by Yesavage et al. (1986), composed of 15 questions [41]. Every question is binary, with a response of “yes”, scoring 0 points, and a response of “no”, scoring 1 point, for a total of 15 points. The higher the value is, the more severe the degree of depression is. The optimal SGDS-K cut-off point varies depending on the study and has previously been established at 6, 8, or 10. The optimal cut-off point for SGDS-K is 8 [42] and, accordingly, 8 points were adopted as the cut-off point in this study as well. The validity and reliability of the Korean translation of SGDS-K are verified in the work of Cho et al. (1999).

Pearson correlation analyses investigating links between the depression scores measured by SGDS-K, nutrition scores measured by NQ or NQ-E, and some of the dietary factors were performed to determine the causes of depression and nutritional status outcomes in dysphagia patients.

### 2.3. Statistical Analysis

All statistical analyses were performed using SPSS software (Statistical Package for the Social Sciences, version 22.0 for Windows). A total of 304 surveys were collected and, following the exclusion of eight surveys due to unreliable or ineligible responses, the analysis was conducted on 141 (47.6%) subjects in the dysphagia group and 155 (52.4%) non-dysphagia group subjects. The total EAT-10 scores were used to classify subjects into the dysphagia or non-dysphagia groups based on a cut-off score of 3 points. For the group comparisons, chi-square tests were performed for the categorical data, including dietary habits, nutritional status, BMI, depression status, and subject characteristics. After deriving the selection attributes of the meal products through exploratory factor analysis, the dysphagia group and the non-dysphagia group were compared by the independent *t*-test. Results were presented as the mean ± standard deviation (SD), frequency, and proportions. Correlation analyses between the total EAT-10 scores, nutritional status, levels of depression, and some dietary factors were performed. Logistic regression analysis was performed to verify the effects of dysphagia on nutritional status and depressive status. In order to analyze the dietary factors affecting the nutritional and depressive status of patients with dysphagia, a correlation analysis was conducted to investigate some of the dietary factors and the NQ and NQ-E scores and depression scores (SGDS-K) using the responses of patients with dysphagia. Lastly, a logistic regression analysis was conducted to investigate links between nutritional status and dietary factors and between depression status and dietary factors using the responses of the dysphagia group in order to verify the effects of some dietary factors on nutrition status and depression status in patients with dysphagia. Significance was set at a level of *p* < 0.05.

## 3. Results

### 3.1. Subject Characteristics

A total of 304 people were recruited and, excluding eight unreliable or ineligible responses, a final sample of responses from 141 dysphagia subjects (47.6%) and 155 normal subjects (52.4%) were considered for the analysis (Figure 1). Demographic characteristics of the survey subjects are presented in Table 1. The average age of the subjects was 73.79 years in the dysphagia group and 70.15 years in the non-dysphagia group, with a total average age of 71.88 years among all subjects. The overall age range of all the subjects was 50 to 92 years. As for sex and marital status, 188 subjects (63.5%) were women and 202 subjects (68.2%) reported having a spouse. One hundred and ninety-six subjects (66.9%) reported that they were not engaged in economic activities, 114 subjects (38.5%) had graduated from middle and high school, 252 subjects (85.1%) were living in their own home, and 60 subjects (20.3%) reported earning an income of less than USD 1000 per month. As a measure for determining dysphagia, the EAT-10 scores were 20.74 ± 11.07 points for the dysphagia group and 0.36 ± 0.65 points for the normal group.

There were significant differences between the dysphagia group and non-dysphagia group in terms of characteristics of age (*p* < 0.001), economic activity (*p* < 0.05), level of education (*p* < 0.001), and housing type (*p* < 0.001). Subjects in the dysphagia group tended to be older than those in the non-dysphagia group, with an age distribution of dysphagia subjects aged 80 years or older (36.2%), subjects in their 70s (32.6%), subjects in their 60s (18.4%), and subjects in their 50s (12.8%), while the normal group showed an age distribution of subjects in their 70s (44.5%), subjects in their 60s (25.2%), subjects in their 50s (16.1%), and subjects in their 80s (14.2%). The rate of economic activity was lower in the dysphagia group (27.0%) than in the normal group (38.7%). As for the levels of education, the distribution in the dysphagia group included middle and high school (32.6%), elementary school (31.9%), a bachelor’s degree (24.8%), junior college (5.7%), and a graduate degree (5.0%), in comparison to the normal group’s distribution of educational levels, comprising middle and high school (43.9%), a bachelor’s degree (27.7%), elementary school (14.2%), a graduate degree (11.6%), and junior college (2.6%). Regarding the housing type, the proportion of people living in welfare facilities was higher in the dysphagia group (28.4%) than in the normal group (1.3%).

### 3.2. Comparison of the Dietary Factors According to Dysphagia

The dietary results according to whether or not patients had dysphagia are presented in Table 2.

There were significant differences in all items among the dietary factors between the dysphagia group and the normal group according to whether or not patients had dysphagia (*p* < 0.001, *p* < 0.01). When measuring whether subjects “need help with meals”, most subjects in the normal group reported being “able to eat without help” (94.8%). When measuring this among subjects in the dysphagia group, however, answers of “need some help” (20.6%) or “always need help or [I am] tube feeding” (14.9%) were reported. When measuring the “amount of food” consumed, subjects in the dysphagia group were found to consume less of their meals and therefore less food overall than subjects in the normal group. The percentage of people with dysphagia responding to items stating that the subject tends to “be full with a few spoonsful” (13.5%) and tends to “be full with a quarter of a bowl” (14.9%) was higher than that in the normal group. The number of subjects that tended to “be full with one bowl” (34.0%) was lower in the dysphagia group than in the normal group. In terms of the duration of the time spent eating a meal, the proportion of subjects reporting “less than 30 min” was 46.8% in the dysphagia group and 66.5% in the non-dysphagia group. Overall, it can be seen that the duration of time spent eating meals by subjects in the dysphagia group was longer than that of the normal group. With multiple respondents reporting their experience of “inconvenience eating”, it was found that people with dysphagia felt uncomfortable at meals for various reasons, in the descending order of “[food being] hard to swallow” (45.7%), “long mealtime” (40.6%), “low diversity of edible foods” (31.9%), “[I] spill a lot” (23.2%), and “cooking is complicated” (15.2%).

The percentage of subjects who reported preparing their meals themselves was higher in the normal group (64.5%) than in the dysphagia group (36.9%). In the dysphagia group, people who prepared meals other than the patients themselves were reported to be “hired caregivers” (27.0%), “spouses” (22.7%), and “son(s) and/or daughter(s) [of patients]” (11.3%)”. The average monthly food purchase costs were USD 160–299 (24.1%) in the dysphagia group and USD 500 or more (23.9%) in the normal group.

### 3.3. Food Product Selection Attributes

Table 3 presents the selection attributes when purchasing food products. The means and standard deviations of each of the 25 selection attributes were obtained, and the validity of the selection attribute questions was verified through exploratory factor analysis. There were no items with a commonality of 0.4 or less, and the exploratory factor analysis was conducted after removing items with a load of 0.5 or less and items with an incorrect load.

Five factors with an eigenvalue of 1 or more were extracted. Factor 1 consisted of “cooking time”, “how to cook”, “product package design”, “easy to open”, “cost”, and “size and amount”, and was named as the “product convenience” factor, with an eigenvalue of 3.680 and explanatory power of 16.001%. Factor 2 consisted of “easy to swallow”, “easy to chew”, “softness”, and “easy to digest” and was named as the “intake” factor, with an eigenvalue of 3.537 and explanatory power of 15.380%. Factor 3 consisted of “ingredient sources”, “sanitation of the manufacturing process”, “type of product”, “product safety”, “brand”, and “taste” and was named as the “menu quality” factor, with an intrinsic value of 3.308 and explanatory power of 14.381%. Factor 4 was composed of “media commercials”, “information from salesman”, “information from hospital”, “reliability of manufacturers”, and “recommendation from acquaintances”, and its intrinsic value was 2.996 and explanatory power was 13.027%. Factor 4 was named as the “marketing” factor. Factor 5 consisted of “related to disease” and “nutritional balance” and was named as the “health/nutritional” factor, with an eigenvalue of 1.940 and explanatory power of 8.436.

The KMO value of the selection attribute factor analysis was 0.897, and Bartlett’s sphericity verification showed that the approximate chi-square value of the selection attribute factor analysis was 3930.918 and the *p* was <0.000; thus, the correlation coefficient between the items was statistically significant. Therefore, it was judged that the factor analysis of the selection attributes of meal products was appropriate.

Table 4 presents the comparison of the results of the selection attribute factors when purchasing meal products according to the presence or absence of a swallowing disorder. In the entire group, it was found that the importance of factors decreased from the “health/nutritional” factor (6.02 ± 1.25) to the “menu quality” factor (5.70 ± 1.13), “intake” factor (5.45 ± 1.51), and “marketing” factor (4.82 ± 1.35). The health/nutritional factor showed the highest score in both the dysphagia group and the non-dysphagia group. Significant differences between the dysphagia group and the non-dysphagia group were found in terms of the “product convenience factor” (*p* < 0.01), “intake factor” (*p* < 0.001), and “menu quality factor” (*p* < 0.05). Product convenience attributes were found to be considered more important in the non-dysphagia group. Each score equated to 4.69 ± 1.22 points for the dysphagia group and 5.15 ± 1.32 points for the non-dysphagia group. As for the intake factor, the dysphagia group showed 5.79 ± 1.19 points and the non-dysphagia group showed 5.14 ± 1.71 points, and the score was higher in the dysphagia group. Finally, the menu quality factor was higher in the normal group, with 5.54 ± 1.03 points in the dysphagia group and 5.85 ± 1.21 points in the non-dysphagia group.

### 3.4. Nutritional Status (NQ/NQ-E)

Significant differences in nutritional status according to age were observed in the dysphagia group (*p* ≤ 0.001). Table 5 shows the subjects’ nutritional status.

The nutritional status of subjects under 65 years of age was expressed as “high”, “medium”, or “low” based on the NQ. For subjects aged 65 or older, the nutritional status was classified as “high”, “medium-high”, “medium-low”, or “low” according to the NQ-E standards, and for the sake of clarity, classifications of “medium-high” and “medium-low” were presented as “medium”. The number of subjects with a nutritional status grade of “high” was significantly less in the dysphagia group (30.5%) than in the non-dysphagia group (47.1%). In contrast, the number of subjects with a nutritional status grade of “low” was significantly higher in the dysphagia group than in the normal group. These results indicate that subjects in the dysphagia group had a higher risk of malnutrition. Table 6 shows that the scores based on the NQ/NQ-E showed significant correlations with the EAT-10 scores (*p* < 0.01). To verify the effect of dysphagia on malnutrition, a logistic regression analysis was performed (Table 7). The independent variable was set as the total score of EAT-10, a tool for screening dysphagia, and the dependent variable, the nutritional status, was classified as “Normal” for the “high” and “medium” grades, and “Malnutrition” for the “low” grade. As a result, the logistic regression model was statistically significant (Hosmer and Lemeshow χ^2^ = 7.721, *p* = 0.172), and the explanatory power of the regression model was 6.1% (Nagelkerke R^2^ = 0.061). It was found that dysphagia (OR = 0.963, *p* < 0.001) had a significant effect on nutritional status. When the degree of dysphagia increases by one unit, the probability of malnutrition increases by 0.963 times.

### 3.5. Body Mass Index (BMI)

In the case of BMI, there was a significant difference between the dysphagia group and the non-dysphagia group (*p* < 0.001). The BMI values of the subjects are presented in Table 5. The dysphagia group had 14.9% underweight, 44.0% normal, 23.4% overweight, and 17.7% obese subjects, while the normal group had 1.3% underweight, 46.5% normal, 27.1% overweight, and 25.2% obese subjects, showing differences in their distribution.

### 3.6. Depression (SGDS-K)

Following the SGDS-K criteria established in the work of Cho (1999), the presence of depression was indicated by SGDS-K scores of 8 points or above (scores below 8 points indicated a normal state or absence of depression). In the group with dysphagia, 53 (37.6%) were depressed and 88 (62.4%) were normal, whereas in the non-dysphagia group, 6 (3.9%) were depressed and 149 (96.1%) were normal (Table 5). There was a significant difference in the depression status between the two groups (*p* < 0.001). The depression statuses of subjects are presented in Table 5, resulting from the correlation analyses between the subjects’ dysphagia scores (EAT-10), NQ/NO-E scores, and SGDS-K scores (*p* < 0.01). The correlation coefficient between EAT-10 and SGDS-K is 0.603 (Table 6), showing that dysphagia has a high correlation with depression. To verify the effect of dysphagia on depression, a logistic regression analysis was performed (Table 7). As for the dependent variable, the depressive state was classified as ‘normal’ for a result of less than 8 points and as ‘depressed’ for a result of more than 8 points. The logistic regression model was statistically significant (Hosmer and Lemeshow χ^2^ = 9.836, *p* = 0.277), and the explanatory power of the regression model was 23.4% (Nagelkerke R^2^ = 0.234). Additionally, it was found that dysphagia (OR = 1.095, *p* < 0.001) had a significant effect on the depressive state. It was found that if the degree of dysphagia increases by one unit, the probability of depression increases by about 1.095 times.

### 3.7. Dietary Factors, Nutritional Status, and Depression in Dysphagia Patients

To analyze the dietary factors that affect the nutritional and depression status of patients with dysphagia, correlations between some of the dietary factors, NQ/NQ-E scores, and SGDS-K scores were analyzed using only the responses of subjects with dysphagia. Dietary factors were analyzed by selecting the isometric scales of the “need for help with meals”, “amount of food consumed at meals”, and “duration of time spent eating a meal” (Table 8 and Table 9). The higher the score was, the higher the subjects’ need for help at meals was. Subjects received points as follows for the responses of “able to eat without help” (1 point), “need some help” (2 points), and “always need help or [patient is] tube feeding” (3 points). Regarding the “amount of food consumed at meals”, the subjects received points as follows for responding that they tended to “be full with a few spoonsful” (1 point), to “be full with a quarter of a bowl” (2 points), to “be full with half of a bowl” (3 points), and to “be full with one bowl” (4 points). Subjects received 5 points for responses of “not full with one bowl” regarding the amount of food consumed at meals. The higher the score was, the more patients ate. The duration of time spent eating meals was calculated as 1 point for “less than 30 min”, 2 points for “30–40 min”, 3 points for “40–50 min”, 4 points for “50 min–1 h”, 5 points for “1–2 h”, and 6 points for “more than 2 h”. The higher the score was, the more time was spent eating meals by the subjects.

Significant correlations were found between the depression scores (SGDS-K), the need for help with meals, and the amount of food consumed during meals (*p* < 0.01). The correlation coefficient between the SGDS-K outcomes indicating depression and the need for help with meals was 0.474. Accordingly, these findings show that dysphagia has a strong correlation with the degree of the need for help with meals. The nutritional scores (NQ/NQ-E) showed a significant correlation with the need for help preparing meals and during meals (*p* < 0.05), but the correlation was not high.

In order to verify the effects of some of the dietary factors of the dysphagia patients on nutrition and depression, logistic regression analysis was performed to investigate links between the nutrition status and dietary factors and the depression status and dietary factors, using only the responses of the dysphagia group. The results are presented in Table 10 and Table 11. The analysis between the nutritional status and dietary factors revealed statistical significance in the logistic regression model (Hosmer and Lemeshow χ^2^ = 4.324, *p* = 0.742), and the explanatory power of the regression model was 7.9% (Nagelkerke R^2^ = 0.079). None of the dietary factors, however, had a significant effect on the nutritional status based on the regression coefficients. In the analysis of the depression status and dietary factors, the logistic regression model was found to be statistically significant (Hosmer and Lemeshow χ^2^ = 4.768, *p* = 0.688), and the explanatory power of the regression model was 31.2% (Nagelkerke R^2^ = 0.312). After testing the significance of the regression coefficients, it was found that the need for help with meals (OR = 3.460, *p* < 0.001) and amount of food consumed at meals (OR = 0.702, *p* < 0.05) had a significant effect on the depression status. It was found that when the need for help with meals increased by one level, the likelihood of depression increased by about 3.460 times, and when the amount of food consumed at meals decreased by one level, the likelihood of depression increased by 0.494. On the other hand, the time of the meal did not have a significant effect on depression.

## 4. Discussion

This study was conducted with the aim of obtaining basic data for identifying ways of improving dysphagia by comparing dietary factors, meal product selection attributes, and nutrition and depression status according to whether adults aged 50 or older have dysphagia and analyzing dietary factors affecting malnutrition and depression in patients with dysphagia. The main findings are as follows. In the case of dietary habits, there was a significant difference between the two groups in all items (*p* < 0.001, *p* < 0.01). The factor analysis of the meal product selection attributes revealed that the product convenience factor, intake factor, menu quality factor, marketing factor, and health/nutritional factor were derived. The intake factor (*p* < 0.001) was significantly higher in the dysphagia group, and the product convenience factor (*p* < 0.01) was significantly higher in the non-dysphagia group. Dysphagia was found to have significant effects on malnutrition (*β* = 0.037, OR = 1.095, *p* < 0.001) and depression (*β* = 0.090, OR = 1.095, *p* < 0.001). Depression in the dysphagia group showed a significantly high correlation with the degree of the need for help with meals and amount of food consumed at meals (*p* < 0.01). The logistic regression analysis found that the dietary factors affecting depression in patients with dysphagia included an increase in the degree of the need for meal help (*β* = 1.241, OR = 3.460, *p* < 0.001) and a decrease in the amount of food consumed at meals (*β* = −0.494, OR = 0.702, *p* < 0.05). These results suggest the possibility that the cause of depression in the elderly with dysphagia is the fact that it is difficult for patients to eat on their own.

The dysphagia group comprised a higher proportion of older subjects and a lower proportion of subjects engaged in economic activity than the non-dysphagia group. Several previous studies have also reported that older age is associated with a higher risk of dysphagia [5,43,44,45,46]. As for the housing type, the proportion of people living in nursing welfare facilities was higher in the dysphagia group than in the normal group. The dysphagia group tended to have a higher need for meal assistance than the normal group, which is supported by previous studies of the elderly in nursing homes [47], in which patients with dysphagia were found to be highly dependent on meal behavior. In the normal group, a majority of subjects reported preparing their own meals, whereas in the dysphagia group, the proportion of other people preparing meals for the dysphagia patients—namely, hired caregivers, spouses, and children (including daughters-in-law)—was high. In terms of purchasing food, subjects in the dysphagia group reported that others purchased food for them at a high rate, including spouses, children, and facility staff, as opposed to purchasing their own food by themselves. The results of the dysphagia group were also compared with those of previous studies [34], which reported that, in general, the elderliness of consumers in Korea has a strong influence on the purchasing and preparing of meals. Based on these results, dysphagia patients are more likely to have food purchased for them by their families, facility staff, and caregivers through shared income than by themselves through their own economic activities. The dietary and nutritional statuses of elderly people are affected by meal preparation and the purchasers of food [48]. Accordingly, it is necessary to establish product and marketing strategies in consideration of the main purchasers and preparers of food in the development of dysphagia-related products in the future.

The meal product selection attributes, which are important for patients with dysphagia, were intake factors. Intake factors derived in this study included “easy to swallow”, “easy to chew”, “softness”, and “easy to digest”, which are consistent with concepts such as “oral comfort” and “oral health” that have been suggested in previous studies [49,50]. “Oral comfort”, presented in a study by Vandenberghe-Descamps et al. (2018), is a new concept of the food sensory characteristics that older consumers experience when eating food. Intra-oral comfort is “due to easy chewing and moistening in the mouth, ease of swallowing and softness of texture when eating food”, which is reduced by pain in the mouth [50]. Therefore, when developing products related to the ease of swallowing in the future, it seems necessary to develop food products that are cooked using wet heat, such as steaming, in consideration of the degree of oral comfort.

Dysphagia is associated with poor nutritional status and depression, and subjects in the dysphagia group of the sample herein were shown to be less healthy in terms of their nutritional status than subjects in the non-dysphagia group. There were differences in BMI between the non-dysphagia group and the dysphagia group, and the results are consistent with malnutrition and intake problems reported in several previous studies [3,14,44,51,52]. Since the problem of swallowing makes the maintenance of proper nutrition difficult by limiting the food intake, patients with swallowing disorders have malnutrition problems [51]. Accordingly, we observed a significant number of dysphagia patients with depression. In addition, dysphagia (EAT-10) was found to have significant effects on malnutrition and depression (*p* < 0.001). Furthermore, the results are consistent with previous studies, showing a positive correlation between dysphagia and depression [53,54]. These results confirm that malnutrition and depression are serious health problems for the elderly with dysphagia.

In this study, depression among subjects in the dysphagia group showed a significant correlation with the degree of help needed by subjects with meals. In addition, the dietary factors affecting depression in patients with swallowing disorders were found to increase the degree of the need for meal assistance and decrease the amount eaten in a meal. According to the above results, restrictions and discomfort during meals, which make it difficult for dysphagia patients to eat by themselves, are reasons for depression in patients with dysphagia. Direct oral intake by patients with dysphagia has a significant effect on life satisfaction [55], and previous studies have also reported that the quality of life of patients with dysphagia improved due to increased dietary levels [56]. The top three uncomfortable eating experiences reported by subjects in the dysphagia group herein were related to food being hard to swallow, meals taking a long time, and low diversity of edible foods. In comparison to the normal group, the dysphagia group reported problems of longer mealtime duration and lesser amounts of food consumed during meals. The issue of small amounts of food consumed at mealtimes is consistent with the findings of a study of community-dwelling elderly individuals in Korea [5]. In particular, low diet variety is associated with malnutrition [57], and malnutrition affects depression and life satisfaction in the elderly [58]. Therefore, it is believed that dysphagia patients require meal products that are easy to eat on their own. These meal products have the potential to increase the dietary variety of the elderly with dysphagia and improve the discomfort they feel when eating.

The results of this study show that decreased self-reliance in the context of eating has the greatest influence on the high depression status of patients with dysphagia. Moreover, the reduced amount of food consumed at meals also has an effect. To improve depression in patients with dysphagia, it is necessary to make it easier for them to eat without help. This will naturally improve their nutritional status by increasing the patients’ food intake. Enriched oral nutritional supplements are ineffective for long-term use because elderly users do not tend to obtain a sufficient intake in order to meet their needs due to the limited taste and texture of supplements [59,60,61,62]. Along these lines, one previous study explored the possibility of meeting the recommended protein intake criteria using familiar foods among elderly eaters [63]. Many successful experiences of increased self-efficacy have been reported [64], and the promotion of the self-efficacy of several patients helped to maintain a healthy diet [65,66]. The development of dietary products, considering the characteristics of dysphagia, food preference, and selection attributes, is expected to offer a way of improving the depression and nutritional status of patients by increasing the intake success of the elderly with dysphagia. Therefore, in order to improve the quality of life and health of an increasing number of middle-aged and older people with dysphagia, emotional support should be provided. Moreover, food development studies reflecting the characteristics of dysphagia patients should be conducted in the future so as to address the root cause of the problem.

This study provided comprehensive information on the nutrition, depression status, and diets of adults aged 50 years or above with dysphagia, and it also derived the cause of depression, a health problem in patients with dysphagia, based on dietary factors. This is of great significance, as there is a lack of prior research on the dietary behaviors of elderly people living with dysphagia. The limitation of this study is that only EAT-10, a self-questionnaire, was used for the classification of the dysphagia group. Therefore, in future studies, it is necessary to classify the dysphagia group using a video fluoroscopic swallowing study (VFSS), which is used to diagnose swallowing disorders in actual clinical practice. In addition, most of the study subjects are residents of Seoul, Korea, which may limit the ability of this cohort to represent the total population of elderly people with dysphagia. A follow-up study considering residential areas and gender ratios is needed. If a follow-up study mitigating the above limitations is conducted in the future, a more in-depth study on the diet of the elderly with dysphagia will be possible.

## 5. Conclusions

This study aims to provide basic data about the ways of improving dysphagia by investigating and comparing the dietary habits, food product selection attributes, and nutritional and depression conditions among adults aged 50 years or above according to whether or not they have dysphagia. The conclusions drawn from the results are as follows. Dysphagia shows a significant correlation with depression and nutritional status, and depression in the elderly with dysphagia was found to be affected by an increase in the degree of the need for help with meals and a decrease in meal size. These findings suggest that it is necessary to develop dietary products that improve certain uncomfortable eating experiences among patients with dysphagia. If discomfort is reduced, then nutritional intake can be increased, and enjoyment in meals can be restored in order to decrease depression in patients with dysphagia. Therefore, in order to improve the quality of life and health of the middle-aged and older people with dysphagia, food development studies reflecting the characteristics of older adults with dysphagia should be conducted.

## Figures and Tables

**Figure 1 nutrients-14-04045-f001:**
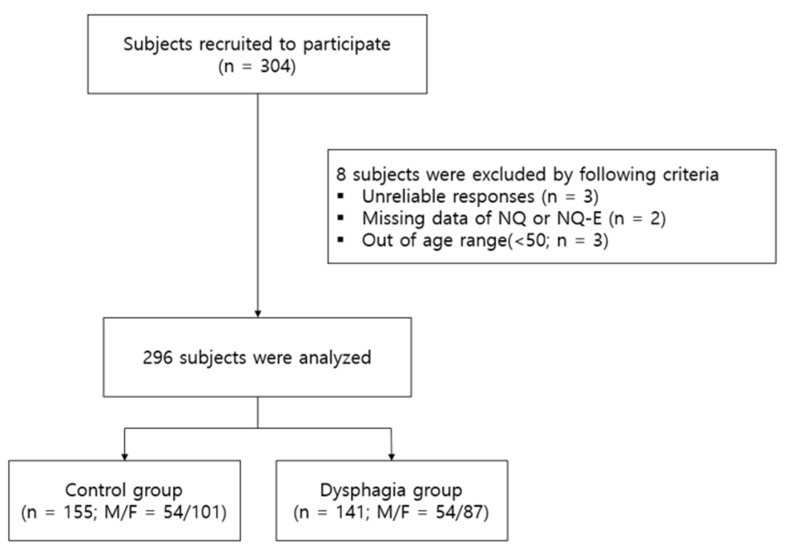
Study flow chart.

**Table 1 nutrients-14-04045-t001:** Subject Characteristics.

		Total	Dysphagia ^4^	Non-Dysphagia	χ^2^
Total		296 (100)	141 (47.6)	155 (52.4)	
Sex	Male	108 (36.5) ^1^	54 (38.3)	54 (34.8)	0.381
	Female	188 (63.5)	87 (61.7)	101 (65.2)	
Age, years	Average	71.88 ± 10.04 ^2^	73.79 ± 10.43	70.15 ± 9.38	-
	50–59	43 (14.5)	18 (12.8)	25 (16.1)	19.241 ***^,3^
	60–69	65 (22.0)	26 (18.4)	39 (25.2)	
	70–79	115 (38.9)	46 (32.6)	69 (44.5)	
	≥80	73 (24.7)	51 (36.2)	22 (14.2)	
Dysphagia	EAT-10	10.07 ± 12.74 ^2^	20.74 ± 11.07	0.36 ± 0.65	-
Marital status	With spouse	202 (68.2)	91 (64.5)	111 (71.6)	1.705
	Without spouse	94 (31.8)	50 (35.5)	44 (28.4)	
Economic activity status	Yes	98 (33.1)	38 (27.0)	60 (38.7)	4.610 *
	No	198 (66.9)	103 (73.0)	95 (61.3)	
Educational level	Elementary school	67 (22.6)	45 (31.9)	22 (14.2)	18.514 ***
	Middle or high school	114 (38.5)	46 (32.6)	68 (43.9)	
	Junior college	12 (4.1)	8 (5.7)	4 (2.6)	
	Bachelor’s degree	78 (26.4)	35 (24.8)	43 (27.7)	
	Graduate degree	25 (8.4)	7 (5.0)	18 (11.6)	
Residence Type	Home	252 (85.1)	100 (70.9)	152 (98.1)	13.533 ***
	Retirement home	1 (0.3)	1 (0.7)	0 (0.0)	
	Welfare facility	42 (14.2)	40 (28.4)	2 (1.3)	
	Other	1 (0.3)	0 (0.0)	1 (0.6)	
Income (per month)	Less than USD 1000	60 (20.3)	30 (21.3)	30 (19.4)	2.648
	USD 1000–USD 1999	55 (18.6)	24 (17.0)	31 (20.0)	
	USD 2000–USD 2999	54 (18.2)	28 (19.9)	26 (16.8)	
	USD 3000–USD 3999	43 (14.5)	18 (12.8)	25 (16.1)	
	USD 4000–USD 4999	30 (10.1)	17 (12.1)	13 (8.4)	
	More than USD 5000	54 (18.2)	24 (17.0)	30 (19.4)	

^1^ N (%); ^2^ mean ± SD; ^3^ * *p* < 0.05, *** *p* < 0.001; ^4^ EAT-10 cut-off: 3 points, possible range of 0–40 points.

**Table 2 nutrients-14-04045-t002:** Dietary Factors in the Dysphagia and Non-Dysphagia Groups.

		Total	Dysphagia	Non-Dysphagia	χ^2^
Total		296 (100.0)	141 (47.6)	155 (52.4)	
Need for help with meals	Able to eat without help	238 (80.4) ^1^	91 (64.5)	147 (94.8)	43.052 ***^,^^2^
	Need some help	34 (11.5)	29 (20.6)	5 (3.2)	
	Always need help or [patient is] tube feeding	24 (8.1)	21 (14.9)	3 (1.9)	
Amount of food consumed at meals	Full with a few spoonsful	22 (7.4)	19 (13.5)	3 (1.9)	22.172 ***
	Full with a quarter of a bowl	33 (11.1)	21 (14.9)	12 (7.7)	
	Full with half of a bowl	112 (37.8)	52 (36.9)	60 (38.7)	
	Full with one bowl	124 (41.9)	48 (34.0)	76 (49.0)	
	Not enough with one bowl	5 (1.7)	1 (0.7)	4 (2.6)	
Duration of mealtimes	Less than 30 min	169 (57.1)	66 (46.8)	103 (66.5)	20.350 ***
	30–40 min	92 (31.1)	48 (34.0)	44 (28.4)	
	40–50 min	18 (6.1)	15 (10.6)	3 (1.9)	
	50 min–1 h	13 (4.4)	8 (5.7)	5 (3.2)	
	1–2 h	4 (1.4)	4 (2.8)	0 (0.0)	
	More than 2 h	0 (0.0)	0 (0.0)	0 (0.0)	
Discomfort while eating ^3^	[Patient] spills a lot	40 (13.9)	32 (23.2)	8 (5.4)	-
	Long mealtime	65 (22.6)	56 (40.6)	9 (6.0)	
	[Food is] hard to swallow	63 (22.0)	63 (45.7)	0 (0.0)	
	Low diversity of edible foods	60 (20.9)	44 (31.9)	16 (10.7)	
	Cooking is complicated	39 (13.6)	21 (15.2)	18 (12.1)	
	No problems	143 (49.8)	32 (23.2)	111 (74.5)	
Person who prepares meals	Myself	152 (51.4)	52 (36.9)	100 (64.5)	51.876 ***
	Spouse	77 (26.0)	32 (22.7)	45 (29.0)	
	Son(s) and/or daughter(s) (including child-in-law)	22 (7.4)	16 (11.3)	6 (3.9)	
	Hired caregiver	42 (14.2)	38 (27.0)	4 (2.6)	
	Other	3 (1.0)	3 (2.1)	0 (0.0)	
Person who buys food	Self	144 (48.6)	48 (34.0)	96 (61.9)	52.953 ***
	Spouse	78 (26.4)	32 (22.7)	46 (29.7)	
	Son(s) and/or daughter(s) (including child-in-law)	39 (13.2)	28 (19.9)	11 (7.1)	
	Welfare facility staff	23 (7.8)	22 (15.6)	1 (0.6)	
	Hired caregiver	10 (3.4)	9 (6.4)	1 (0.6)	
	Other	2 (0.7)	2 (1.4)	0 (0.0)	
Purchasing cost of food (per month)	Less than USD 50	11 (3.7)	3 (2.1)	8 (5.2)	16.617 **
	USD 50–USD 159	48 (16.2)	14 (9.9)	34 (21.9)	
	USD 160–USD 299	60 (20.3)	34 (24.1)	26 (16.8)	
	USD 300–USD 500	67 (22.6)	31 (22.0)	36 (23.2)	
	More than USD 500	68 (23.0)	31 (22.0)	37 (23.9)	
	Patient does not know	42 (14.2)	28 (19.9)	14 (9.0)	

^1^ N (%), ^2^ ** *p* < 0.01, *** *p* < 0.001, ^3^ multiple responses.

**Table 3 nutrients-14-04045-t003:** Factor analysis of the selection attributes when purchasing food products.

							*n* = 296
Factor	Variance	Mean ± SD	Factor Loading	Commonality	Eigenvalue	Persuasive Power of Variance (%)	Cronbach’s α
Product Convenience	Cooking time	4.90 ± 1.59	0.811	0.715	3.680	16.001	0.855
	How to cook	5.15 ± 1.68	0.777	0.703			
	Product package design	3.85 ± 1.85	0.728	0.701			
	Easy to open	5.02 ± 1.84	0.638	0.584			
	Cost	5.54 ± 1.60	0.622	0.546			
	Size and amount	5.11 ± 1.61	0.611	0.503			
Intake	Easy to swallow	5.40 ± 1.77	0.900	0.879	3.537	15.380	0.923
	Easy to chew	5.22 ± 1.76	0.883	0.842			
	Softness	5.35 ± 1.61	0.824	0.765			
	Easy to digest	5.83 ± 1.57	0.808	0.776			
Menu Quality	Ingredient sources	5.82 ± 1.55	0.724	0.627	3.308	14.381	0.829
	Sanitation of the manufacture process	6.14 ± 1.38	0.714	0.631			
	Type of product	5.32 ± 1.57	0.681	0.618			
	Product safety	6.30 ± 1.33	0.675	0.698			
	Brand	4.78 ± 1.83	0.569	0.624			
	Taste	5.84 ± 1.54	0.528	0.461			
Marketing	Media commercial	4.22 ± 1.83	0.805	0.749	2.996	13.027	0.844
	Information from salesman	4.26 ± 1.88	0.796	0.688			
	Information from hospital	5.26 ± 1.66	0.681	0.726			
	Reliability of manufacturers	5.38 ± 1.58	0.649	0.694			
	Recommendation from acquaintances	4.98 ± 1.67	0.631	0.615			
Health/nutritional	Related to disease	5.79 ± 1.63	0.673	0.655	1.940	8.436	0.687
	Nutritional balance	6.25 ± 1.21	0.645	0.662			

Total variance = 67.226, Kaiser–Meyer–Olkin measure of sampling adequacy (KMO) = 0.897, Bartlett’s test of sphericity, chi-square = 3930.918 (df = 253, sig. = 0.000).

**Table 4 nutrients-14-04045-t004:** Attributes of the selection of food products in the dysphagia and non-dysphagia groups.

				Mean ± SD
Factor	Total (*n* = 296)	Dysphagia (*n* = 141)	Non-Dysphagia (*n* = 155)	*t*-Value
Product Convenience	4.93 ± 1.29	4.69 ± 1.22	5.15 ± 1.32	−3.106 **^,1^
Intake	5.45 ± 1.51	5.79 ± 1.19	5.14 ± 1.71	3.783 ***
Menu Quality	5.70 ± 1.13	5.54 ± 1.03	5.85 ± 1.21	−2.384 *
Marketing	4.82 ± 1.35	4.68 ± 1.31	4.95 ± 1.39	−1.745
Health/Nutritional	6.02 ± 1.25	6.02 ± 1.09	6.02 ± 1.38	0.038

Five-point Likert scale: 5 = highest, 1 = lowest; ^1^ * *p* < 0.05, ** *p* < 0.01, *** *p* < 0.001.

**Table 5 nutrients-14-04045-t005:** Body Mass Index, Nutritional Status, and Depression Status by Group.

		Total	Dysphagia	Non-Dysphagia	χ^2^
**Total**		296 (100)	141 (47.6)	155 (52.4)	
Body mass index ^4^	Underweight	23 (7.8)	21 (14.9)	2 (1.3)	19.96 ***^,2^
	Normal	134 (45.3)	62 (44.0)	72 (46.5)	
	Overweight	75 (25.3)	33 (23.4)	42 (27.1)	
	Obese	64 (21.6)	25 (17.7)	39 (25.2)	
Nutritional status	High	116 (39.2) ^1^	43 (30.5)	73 (47.1)	13.533 ***
	Medium	123 (41.6)	60 (42.6)	63 (40.6)	
	Low	57 (19.3)	38 (27.0)	19 (12.3)	
Depression status ^3^	Depressed	59 (19.9)	53 (37.6)	6 (3.9)	52.597 ***^,3^
	Normal	237 (80.1)	88 (62.4)	149 (96.1)	

^1^ N (%), ^2^ *** *p* ≤ 0.001, ^3^ SGDS-K cut-off: 8 points; ^4^ underweight: BMI < 18.5, normal: 18.5 ≤ BMI < 23, overweight: 23 ≤ BMI < 25, obese: 25 < BMI.

**Table 6 nutrients-14-04045-t006:** Correlations between the 10-item Eating Assessment Tool, Nutrition Quotient/Nutrition Quotient for the Elderly, and the Short-Form Geriatric Depression Scale for Koreans.

Variables	Mean ± SD	Inter-Construct Correlations		
		1	2	3
1. EAT-10	10.07 ± 12.74	1		
2. SGDS-K	3.73 ± 4.23	0.603 **	1	
3. NQ or NQ-E	60.03 ± 11.37	−0.217 **	−0.338 **	1

** *p* < 0.01.

**Table 7 nutrients-14-04045-t007:** Correlations between Dysphagia (EAT-10) and Malnutrition and Depression using Logistic Regression.

Variable	*β*	S.E.	OR	95% CI	*p*
Malnutrition	0.037	0.011	0.963 ***	(0.943~0.984)	0.001
	−2LL = 278.477, Nagelkerke R^2^ = 0.61, Hosmer and Lemeshow test: χ^2^ = 7.721 (*p* = 0.172)				
Depression	0.090	0.019	1.095 ***	(1.054~1.137)	0.000
	−2LL = 160.099, Nagelkerke R^2^ = 0.234, Hosmer and Lemeshow test: χ^2^ = 9.836 (*p* = 0.277) Cut-off: 8 point				

*** *p* < 0.001.

**Table 8 nutrients-14-04045-t008:** Correlation between the Short-Form Geriatric Depression Scale for Koreans and Dietary Factors in the Dysphagia Group.

Variables	Mean ± SD	Inter-Construct Correlations			
		1	2	3	4
1. SGDS-K	5.94 ± 4.77	1			
2. Need for help with meals	1.50 ± 0.74	0.474 **	1		
3. Duration of time spent at meals	1.84 ± 1.02	0.005	0.072	1	
4. Amount of food consumed at meals	2.94 ± 1.03	−0.302 **	−0.313 **	−0.126	1

** *p* < 0.01.

**Table 9 nutrients-14-04045-t009:** Correlation between Nutrition Quotient/Nutrition Quotient for the Elderly Scores and Dietary Factors in Dysphagia.

Variables	Mean ± SD	Inter-Construct Correlations			
		1	2	3	4
1. NQ or NQ-E	57.51 ± 10.34	1			
2. Need for help with meals	1.50 ± 0.74	−0.201 *	1		
3. Duration of time spent at meals	1.84 ± 1.02	0.196 *	0.072	1	
4. Amount of food consumed at meals	2.94 ± 1.03	−0.048	−0.313 **	−0.126	1

* *p* < 0.05, ** *p* < 0.01.

**Table 10 nutrients-14-04045-t010:** Odds ratio estimates of the need for help with meals, amount of food consumed at meals, duration of time spent at meals, and nutritional status.

Variable	*β*	S.E.	OR	95% CI	*p*
Need for help with meals	0.320	0.262	1.377	(0.823~2.302)	0.223
Amount of food consumed at meals	−0.354	0.195	0.702	(0.479~1.028)	0.069
Duration of time spent at meals	−0.302	0.215	0.740	(0.485~1.128)	0.161
−2LL = 156.441, Nagelkerke R^2^ = 0.079, Hosmer and Lemeshow test: χ^2^ = 4.324 (*p* = 0.742)					

**Table 11 nutrients-14-04045-t011:** Odds ratio estimates of the need for help with meals, amount of food consumed at meals, duration of time spent at meals, and depression status.

Variable	*β*	S.E.	OR	95% CI	*p*
Need for help with meals	1.241	0.288	3.460 ***	(1.968~6.082)	0.000
Amount of food consumed at meals	−0.494	0.202	0.702 **	(0.411~0.906)	0.014
Duration of time spent at meals	−0.078	0.202	0.740	(0.623~1.373)	0.698
−2LL = 150.049, Nagelkerke R^2^ = 0.312, Hosmer and Lemeshow test: χ^2^ = 4.768 (*p* = 0.688),cut-off: 8 points					

** *p* < 0.01, *** *p* < 0.01.

## Data Availability

Not applicable.

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
