# Peer review of "Dietary Habits, Food Product Selection Attributes, Nutritional Status, and Depression in Middle-Aged and Older Adults with Dysphagia"

_nutrients, 2022, doi:10.3390/nu14194045_

Round 1

Reviewer 1 Report

The study aims to understand the dietary habits of the middle-aged and the selection attributes of meal products, the current nutritional status, and the degree of depression according to the presence or absence of dysphagia. The results confirm that malnutrition and depression are serious health problems for the elderly with  dysphagia.

 In the "Introduction" section, insert the references n.12, n.16, n. 21, n. 22, n. 26, n. 27.

 Results

line 190: the value 66.7% does not correspond to Tab 1

line 204: the value 27.3% does not correspond to Tab 1

line 210: the value 28.8% does not correspond to Tab 1

line 242: the 42.1% value does not correspond to Tab 2

In  table 2: “purchasing cost of food”: who has a value of $ 300, in which category is he placed? 160-300 $ or 300-500 $? Perhaps $ 160-299 should be corrected.

Tables 6 and 7 are presented twice.

Author Response

Response to Reviewer 1 Comments

Thanks for your kind and detailed comments. I revised the manuscript as you commented, which made it better. The responses to the comment were highlighted in red.

-

The study aims to understand the dietary habits of the middle-aged and the selection attributes of meal products, the current nutritional status, and the degree of depression according to the presence or absence of dysphagia. The results confirm that malnutrition and depression are serious health problems for the elderly with dysphagia.

-  Thanks for your comments. I am really glad to get review comments of this manuscript from you.

Point 1: In the "Introduction" section, insert the references n.12, n.16, n. 21, n. 22, n. 26, n. 27.

Response 1: I insert the references n.12(Line 39), n.16(Line 41), n.21(Line 47), n.22(Line 47), n.26(Line 48), n.27(Line 48). Thanks for your comments.

Point 2: Results

line 190: the value 66.7% does not correspond to Tab 1

line 204: the value 27.3% does not correspond to Tab 1

line 210: the value 28.8% does not correspond to Tab 1

line 242: the 42.1% value does not correspond to Tab 2

Response 2: I changed the above figures to the correct ones. Thanks for your detailed comments.

(Line 190) 66.7% -> 66.9%

(Line 204) 27.3% -> 27.0%

(Line 210) 28.8% -> 28.4%

(Line 242) 42.1% -> 24.1%

Point 3: In  table 2: “purchasing cost of food”: who has a value of $ 300, in which category is he placed? 160-300 $ or 300-500 $? Perhaps $ 160-299 should be corrected.

Response 3: I revised “$160-$300” to “$160-$299” in table 2. Thank you.

(Table 2) “$160-$300” -> “$160-$299”

Point 4: Tables 6 and 7 are presented twice.

Response 4: As you commented, Tables 6 and 7 are overlapped, so I corrected the incorrectly marked table number from Table 8. Thanks for your comments.

(Line 368) Table 6 -> Table 8

(Line 370) Table 7 -> Table 9

(Line 390) Table 8 -> Table 10

(Lin3 392) Table 9 -> Table 11

-

Thank you for reviewing this manuscript and writing your valuable comments.

Reviewer 2 Report

This paper investigated dietary habits, food product selection, nutritional and depression status in subjects aged 50 years or older, and compared them between dysphagic and normal subject groups. The authors recommend a necessity of mental support in treating aged dysphagic patients. The study concept is interesting and the study is well designed. Also, the results are attractive.

My just one comment is that 42.1% in line 242 should be 24.1% according to Table 2.

Author Response

Response to Reviewer 2 Comments

Thanks for your kind and detailed comments. I revised the manuscript as you commented, which made it better. The responses to the comment were highlighted in red.

-

This paper investigated dietary habits, food product selection, nutritional and depression status in subjects aged 50 years or older, and compared them between dysphagic and normal subject groups. The authors recommend a necessity of mental support in treating aged dysphagic patients. The study concept is interesting and the study is well designed. Also, the results are attractive.

-  Thanks for your comments. I am really glad to get review comments of this manuscript from you.

Point 1: My just one comment is that 42.1% in line 242 should be 24.1% according to Table 2.

 Response 1: I changed the above figures to the correct ones. Thanks for your detailed comments.

(Line 242) 42.1% -> 24.1%

-

Thank you for reviewing this manuscript and writing your valuable comments.
